# Time-Dependent Changes in the Intestinal Microbiome of Gilts Exposed to Low Zearalenone Doses

**DOI:** 10.3390/toxins11050296

**Published:** 2019-05-24

**Authors:** Katarzyna Cieplińska, Magdalena Gajęcka, Michał Dąbrowski, Anna Rykaczewska, Sylwia Lisieska-Żołnierczyk, Maria Bulińska, Łukasz Zielonka, Maciej T. Gajęcki

**Affiliations:** 1Microbiology Laboratory, Non-Public Health Care Centre, Limanowskiego 31A, 10-342 Olsztyn, Poland; kasiacieplinska@gmail.com; 2Department of Veterinary Prevention and Feed Hygiene, Faculty of Veterinary Medicine, University of Warmia and Mazury in Olsztyn, Oczapowskiego 13, 10-718 Olsztyn, Poland; michal.dabrowski@uwm.edu.pl (M.D.); anna.rykaczewska@uwm.edu.pl (A.R.); lukasz.zielonka@uwm.edu.pl (Ł.Z.); gajecki@uwm.edu.pl (M.T.G.); 3Independent Public Health Care Centre of the Ministry of the Interior and Administration, and the Warmia and Mazury Oncology Centre in Olsztyn, Wojska Polskiego 37, 10-228 Olsztyn, Poland; lisieska@wp.pl; 4Department of Discrete Mathematics and Theoretical Computer Science, Faculty of Mathematics and Computer Science, University of Warmia and Mazury in Olsztyn, Słoneczna 34, 10-710 Olsztyn, Poland; bulma@uwm.edu.pl

**Keywords:** zearalenone, doses, intestinal microbiome, intestinal mycobiome, pre-pubertal gilts

## Abstract

Zearalenone is a frequent contaminant of cereals and their by-products in regions with a temperate climate. This toxic molecule is produced naturally by *Fusarium* fungi in crops. The aim of this study was to determine the influence of low zearalenone doses (LOAEL, NOAEL and MABEL) on the intestinal microbiome of gilts on different days of exposure (days 7, 21 and 42). Intestinal contents were sampled from the duodenal cap, the third part of the duodenum, jejunum, caecum and the descending colon. The experiment was performed on 60 clinically healthy gilts with average BW of 14.5 ± 2 kg, divided into three experimental groups and a control group. Group ZEN5 animals were orally administered ZEN at 5 μg /kg BW, group ZEN10—10 μg ZEN/kg BW and group ZEN15—15 µg ZEN/kg BW. Five gilts from every group were euthanized on analytical dates 1, 2 and 3. Differences in the log values of microbial counts, mainly *Escherichia coli* and *Enterococcus faecalis*, were observed between the proximal and distal segments of the intestinal tract on different analytical dates as well as in the entire intestinal tract. Zearalenone affected the colony counts of intestinal microbiota rather than microbiome diversity, and its effect was greatest in groups ZEN10 and ZEN15. Microbial colony counts were similar in groups ZEN5 and C. In the analysed mycobiome, ZEN exerted a stimulatory effect on the log values of yeast and mould counts in all intestinal segments, in particular in the colon, and the greatest increase was noted on the first analytical date.

## 1. Introduction

Plant materials and their by-products are used in feed production, which increases the risk of mycotoxin (undesirable substance) poisoning in humans [1] and livestock, pigs in particular. Exposure to high doses of mycotoxins, including zearalenone (ZEN), has been well documented [2,3]. However, extensive research conducted in the last decade indicates that health problems resulting from exposure to small doses of the parental compound [4,5,6] without modified mycotoxins [7] can be equally important. The above is confirmed by the hormesis paradigm [7,8]. Doses below LOAEL values (lowest observed adverse effect level) [9,10,11] which induce pathological changes without clinical symptoms (sub-clinical states) are referred to as NOAEL doses (no observed adverse effect level) [12].

The minimal anticipated biological effect level (MABEL) dose enters into positive interactions with macroorganisms in different stages of their life cycle [13]. However, this observation contradicts the low-dose hypothesis, which plays an important role in relation to natural and hormonally active compounds [14] such as ZEN. In animals, the variations in the dose-response relationship induce differences in the interpretation of clinical symptoms and laboratory tests evaluating the risk of contamination with low doses of mycosteroids, such as ZEN, in plant materials [15]. In biomedical practice, an accurate determination of low mycotoxin doses in plant material would support a more reliable interpretation of the final effects [16].

Our findings indicate that the duodenum and the jejunum play the most important role in ZEN absorption [17,18]. The above can be attributed to the anatomical structure of the proximal segments of the small intestine, in particular differences in the quality and quantity of mucus glycoproteins. The discussed segments of the small intestine are characterized by very small quantities of strongly adhering mucus; therefore, digested nutrients have easy access to the intestinal wall [19]. Dietary sources of energy are also highly available in the proximal segments of the small intestine [20]. In animals exposed to mycotoxins, the energy derived from the diet promotes biotransformation processes that are essential for porcine health [13]. In the first week of exposure, a physiological deficiency of endogenous steroids inhibits biotransformation, and mycosteroids become deposited in intestinal tissues only in successive weeks [10,21,22].

Initially, ZEN is accumulated mainly in the duodenum. The above is observed until the end of the third week of exposure to the parental compound. In the fifth week of exposure, the accumulation of ZEN was highest in the descending colon [13].

The changes induced by exposure to low mycotoxin doses have been insufficiently investigated in the literature. In view of the hormesis paradigm, the variations in clinical symptoms or the absence of such symptoms lead to doubt in clinical evaluations. These doubts result not only from the dose, but also from the time of exposure [23]. There are three possible causes of the above. The first is the body’s failure to recognise the threat [24], which is consistent with the T-regs theory [25]. The second is the compensatory effect, namely increased absorption to compensate for the physiological deficiency of endogenous steroids [26,27].

A review of the literature indicates that diet also influences the type and severity of physiological responses to ZEN in the porcine digesta. Diet and exposure to ZEN determine the specific composition of intestinal microbiome [28]. According to some researchers, due to its ecological complexity, the microbiome should be regarded as a “microbiological organ” which enters into dynamic interactions with the host and digesta [29] throughout the host’s life. Intestinal microbiome stimulates the production of vitamins and cofactors, enhances digestion, eliminates feed toxins, creates an inner microbiological layer in the intestine that physically removes pathogens, produces natural antibiotics and fungicidal compounds, maintains intestinal barrier function and promotes the anti-inflammatory response [30]. Microbiome composition significantly affects gut health, nutrient utilization and bodily functions in pigs [31]. For this reason, the presence of ZEN in digesta can induce changes in ecological homeostasis and lead to dysbiosis in gut microbiota [4,32]. The above can promote local adhesion of pathogenic bacteria and the development of intestinal inflammations [33,34].

A well-balanced gut microbiome with a stable qualitative and quantitative composition is required for healthy bodily function in animals [23,35]. Intestinal bacteria stimulate the immune response and produce metabolites which are important for the host’s well-being. Gut microflora facilitate nutrient absorption, deliver protective effects, stimulate the immune system, promote fermentation processes and prevent pathogen colonization. Intestinal bacteria are also used in the prevention and treatment of inflammatory bowel diseases (IBD) [36].

Despite those benefits, microorganisms can also exert negative effects on animals [37]. Pathogens produce toxic metabolites and faecal enzymes which can promote the generation of carcinogenic substances [38,39]. In the literature, the influence of gut bacteria has been analysed mainly in the context of intestinal microflora’s ability to remove mycotoxins. The mechanisms by which ZEN can induce quantitative changes in bacterial microflora have not been fully elucidated [4,32,40,41].

The existing knowledge about the gut microbiome has been derived mainly from analyses of the isolated microorganisms and their phenotypic identification. Genotyping methods, including analyses of the highly conserved regions of the 16S rRNA gene, revealed that 20% to 60% of microorganisms (gut microbiome) cannot be cultured in vitro. Genotyping also demonstrated that the qualitative composition of the gut microbiome is far more complex and individually varied than initially believed. The gut microbiome is modified by age, environment, diet/feed type, genetic factors, animal welfare standards and the presence of undesirable substances. In human medicine, newly identified sequences of the 16S rRNA gene from various ecological niches were compared with known sequences to detect and identify (to the species level) microorganisms that cannot be cultured or are difficult to culture in vitro. However, the properties and functions of the bacteria identified in a given ecosystem are not always easy to determine. Genetic analyses are carried out to elucidate the potential roles of such microorganisms. Metagenomics (population genomics of environmental microorganisms) tools have been used to analyse the collective genome of gut microbiota based on DNA acquired directly from environmental samples [41,42]. In contemporary research, the aim of quantitative and qualitative analyses is not only to identify the sequences of the 16S rRNA gene, but also to identify the genes encoding specific traits and to generate comprehensive information about the gut microbiome. The most popular analytical techniques are denaturing gradient gel electrophoresis, real-time PCR, microarray analyses, cloning and sequencing, including pyrosequencing [29]. However, phenotypic identification of microbiota produces more comprehensive results that are easier to apply in clinical practice.

The aim of this study was to determine the effect (dose, variability of microbiota) of low doses of ZEN (MABEL, NOAEL and LOAEL) on microbial counts in the porcine gut on different days of exposure with the use of conventional analytical methods.

## 2. Results and Discussion

### 2.1. Experimental Feed

The analysed feed did not contain mycotoxins, or its mycotoxin content was below the sensitivity of the method (VBS). The concentrations of modified mycotoxins were not analysed [5,6].

### 2.2. Clinical Observations

Clinical signs of ZEN mycotoxicosis were not observed throughout the experiment. However, changes in specific tissues or cells were frequently noted in analyses of selected biochemical parameters in samples collected from the same animals and in those animals’ growth performance. The results of these analyses were published in a different paper [5,6].

### 2.3. Evaluation of the Gut Microbiome

#### 2.3.1. General Information

The evaluated microbiota was discussed in the following order: Enterobacteriaceae—Escherichia, Citrobacter, Salmonella, Yersinia and Klebsiella; Enterococcaceae—Enterococcus; Staphylococcaceae—Staphylococcus; Clostridiaceae—Clostridium; Incertae sedis—Candida; Nectriaceae—Fusarium.

Bacteria of the genera *Citrobacter*, *Salmonella*, *Klebsiella* and *Yersinia* were not identified in microbiological analyses on date D1.

Bacteria of the genera *Citrobacter* ad *Salmonella* were not detected on date D2. In group ZEN15, bacteria of the genus *Yersinia* were detected at 0.8 to 6.0 log CFU/g in the third part of the duodenum and the caecum, respectively. In group ZEN5, bacteria of the genus *Klebsiella* were identified in all evaluated intestinal segments at 0.4, 0.4, 3.4, 9.0 and 3.0 log CFU/g, respectively. *Klebsiella* pathogens were detected at 3.0 log CFU/g in the caecum in group ZEN10, and at 9.0 and 3.0 log CFU/g in the caecum and the descending colon, respectively, in group ZEN15.

Bacteria of the genera *Citrobacter*, *Salmonella* and *Yersinia* were not identified on date D3. *Klebsiella* pathogens colonised the third part of the duodenum at 3.0 log CFU/g in group ZEN10 and successive intestinal segments (6.0, 3.0, 6.0 and 6.0 log CFU/g), excluding the descending colon, in group ZEN15.

The overgrowth of the small intestinal microbiome, including changes in microbial counts and/or microbial types, was not observed. In the proximal segment of the small intestine, the counts of non-pathogenic bacterial strains exceeded 105 log CFU due to colonisation by bacterial strains that are ubiquitous in the colon. Only the results where significant differences were noted are presented in the figures.

Significant differences were not observed in the presented results.

#### 2.3.2. Microbiome Analysis in Different Intestinal Segments (Dose Effect)

This subsection analyses the effects of the applied ZEN doses on quantitative changes in microbiota in different groups on different analytical dates, in the same segment of intestinal tract.

##### Duodenal Cap

The duodenal cap is the proximal segment of the duodenum which ends in the major duodenal papilla [43]. This segment has a specific anatomical structure, and in some respects, it resembles the stomach more than the intestine [44]. The duodenal cap receives blood from two different sources. In pre-pubertal gilts, the muscular layer is not yet fully developed, which can lead to the retention of digesta. The local microbiome is not highly diverse due to gastric acid secretions. In the experiment, the counts of *Enterococcus faecalis* were fairly stable at up to 18 log CFU/g throughout the experiment (see Figure 1A). However, considerable differences were observed between groups (5 to 18 log CFU/g) on selected dates.

Variations were also noted in the counts of coagulase-negative *Staphylococci* (see Figure 1B). The highest microbial counts (log CFU/g) on selected analytical dates were found in group ZEN5. *Enterococcus faecalis* are ubiquitous in the gastrointestinal tract, and they maintain intestinal homeostasis [31], excluding selected pathogenic strains which are classified based on their virulence [45]. These strains are conditional pathogens [46]. In this experiment, bacterial counts were inversely proportional to ZEN dose (see Figure 1). Coagulase-negative *Staphylococci* occupy an ecological niche in animal farms, and their pathogenic effects have not yet been fully investigated [47]. These opportunistic bacteria participate in endogenous infections [48].

##### Third Part of the Duodenum

A significant decrease in the counts (log CFU/g) of *Escherichia coli* (see Figure 2A), *Enterococcus faecalis* (see Figure 2B) and coagulase-negative *Staphylococci* (see Figure 2C) was noted in groups ZEN10 and ZEN15 on all analytical dates relative to group C. The differences in the counts of coagulase-negative *Staphylococci* were highly significant on date D1. An analysis of differences in bacterial counts (log CFU/g) between analytical dates revealed the greatest variations in group ZEN5 in the mean counts of *Enterococcus faecalis* and coagulase-negative *Staphylococci* (3 to 16 CFU/g and 15 to 19 log CFU/g, respectively; see Figure 3C,D) on date D1 in the mean counts of *Escherichia coli* (26 to 43 log CFU/g; see Figure 2B) on date D3.

In group C, the average counts of coagulase-negative *Staphylococci* (15 and 13 log CFU/g) on date D1 also differed most significantly (by 15 and 13 log CFU/g) relative to the remaining analytical dates. Highly similar results were reported in a study where antibiotics were used as growth promoters in pigs [49]. Sub-therapeutic doses of antibiotics induced similar changes in the intestinal ecosystem [30] to the NOAEL doses of ZEN (group ZEN10) in this experiment. In turn, MABEL doses (group ZEN5) increased the counts of saprophytic bacteria which inhibited the adhesion of pathogenic cells to the intestinal epithelium. Therefore, a question arises whether low doses of ZEN can induce eubiotic effects [50].

On date D2, bacterial counts (log CFU/g) in the duodenum decreased relative to dates D1 and D3. The average differences in the abundance of *Enterococcus faecalis* ranged from 15 to 18 log CFU/g in group ZEN5 in the duodenal cap (see Figure 1A,B) and the third part of the duodenum (*Escherichia coli* was not identified in the duodenum in group ZEN15 on date D2). The average difference in *Escherichia coli* counts was estimated at 44 log CFU/g in group ZEN5 (see Figure 2A). The above observations validate our previous findings [4] as well as the results presented by Gajęcka et al. [10] which indicate that ZEN has bacteriostatic or even bactericidal effects [49].

##### Jejunum

Similar results were noted in the jejunum. On date D2 (see Figure 3A), *Escherichia coli* counts (log CFU/g) decreased in all groups, and significant differences were observed only between dates in group C. The decrease in *Escherichia coli* counts was directly proportional to the increase in ZEN dose (from 34 and 30 log CFU/g to 13 and 12 log CFU/g, respectively), and significant differences were observed between group C vs. groups ZEN10 and ZEN 15 (see Figure 3A).

Differences were also observed in the counts (log CFU/g) of *Enterococcus faecalis* (Figure 3B) and coagulase-negative *Staphylococci* (see Figure 3C). Coagulase-negative *Staphylococci* were not detected on date D2 (see Figure 3C). Considerable functional variations were noted in *Clostridium difficile* (see Figure 3D) on all dates and in all groups. These findings suggest that ZEN decreases the counts of mesophilic aerobes [49], *Clostridium difficile* (obligate anaerobes), *Escherichia coli* and other bacteria of the family *Staphyococaceae* during and after 42 days of exposure. The above could indicate that prolonged exposure to LOAEL (group ZEN15) or NOAEL (group ZEN10) doses eliminates bacteria or significantly decreases their counts. Similar results were reported by Piotrowska et al. [4] where the mycotoxin dose was 40 ug/kg BW. Bacterial abundance (log CFU/g) was maintained at a higher level only under exposure to the MABEL dose (group ZEN5). The above could suggest that high doses of ZEN exert bacteriostatic or bactericidal effects [49], whereas the lowest dose of ZEN has stimulating properties [10]. Similar results were observed in a study evaluating the genotoxicity of caecal water in the same gilts [5] where genotoxic processes of various intensity were noted in groups ZEN10 and ZEN15. Genotoxicity was not reported in group ZEN5. It should be noted, however, that ZEN is absorbed mainly in the duodenum (65%) [51]. In hypoestrogenic gilts, ZEN is directly used in steroidogenesis or is converted to α-zearalenol [10]. The resulting modified mycotoxin [7,51] is more toxic and/or more metabolically active [6], depending on the dose of the parental compound.

##### Caecum

The following sampled segment of the intestinal tract was the caecum. In pre-pubertal gilts, the caecum is not yet fully developed, and intestinal dysfunctions often originate in this segment of the digestive system [29]. Resistant starch (RS) is not degraded by digestive enzymes in proximal segments of the intestinal tract, and it can be fermented by residual microbiota in the colon [30]. *Enterobacteria* and other bacterial species decompose RS into short-chain fatty acids (SCFAs) which promote the proliferation of caecal cells, increase the expression of genes that participate in intestinal development and acidify the local ecosystem [52]. An acidic environment inhibits the growth of pathogenic microbiota and selectively promotes the growth of selected beneficial microorganisms. Therefore, RS contributes to intestinal health by modifying and stabilizing the populations of intestinal microorganisms and boosting immunity [53]. At the same time, exposure to ZEN inhibits the production of SCFAs [49]. These observations explain the absence of bacteria of the genera *Citrobacter*, *Salmonella*, *Klebsiella* and *Yersinia* and selected coagulase-negative *Staphylococci* on dates D1-D3 (see Figure 4A). In turn, *Escherichia coli* and *Enterococcus faecalis* play a less important role in the fermentation process, which is why their counts (log CFU/g) were similar in the experimental groups and in group C in all intestinal segments [54].

An analysis of Figure 4A,B indicates that bacterial counts (log CFU/g) in the caecum were lower in the experimental groups than in group C on day D1 (see Figure 4A,B). Coagulase-negative *Staphylococci* were not detected on successive sampling dates (D2 and D3) (see Figure 4A), which could be attributed to ongoing fermentation processes in the caecum [31] and exposure to ZEN. The observed processes create a closed-loop system: ZEN decreases microbial counts, which inhibits SCFA synthesis and, consequently, leads to dysbiosis in the caecum [5,55].

On date D1, the counts (log CFU/g) of coagulase-negative *Staphylococci* and *Clostridium difficile* were higher in group ZEN5 than in groups ZEN10, ZEN15 and C, but the observed differences were not statistically significant (see Figure 4A,B). Coagulase-negative *Staphylococci* (see Figure 4A) were not detected on dates D2 and D3 (see Figure 7A,B). This is a desirable situation from the perspective of animal health, but it should be noted that these pathogens are frequently undetected in laboratory analyses [56,57]. The population of *Clostridium difficile* (see Figure 4B) increased proportionally with a rise in ZEN dose. These observations suggest that LOAEL and NOAEL doses of ZEN contribute to subclinical pathological states, in particular those caused by opportunistic strains such as *Clostridium difficile* [58]. This bacterial strain is responsible for intestinal inflammations [59] in piglets and grower-finishing pigs and causes substantial losses in commercial farms [60,61].

##### Descending Colon

The descending colon was the last analysed segment of the intestinal tract. The presented values of $\bar{x}$ and SD indicate that the counts (log CFU/g) of *Escherichia coli* (see Figure 5A) and coagulase-negative *Staphylococci* (see Figure 5B) decreased over time. The results are nearly identical to those noted in the caecum. Significant differences (*p* ≤ 0.05) in *Escherichia coli* counts (log CFU/g) were observed between group ZEN5 and group ZEN10 and between group C and group ZEN10 on date D1, and between group ZEN5 and groups ZEN10 and ZEN15 on date D3 (35, 36, 29.6 and 26.6 log CFU/g, respectively). *Staphylococcaceae* counts differed between group C and group ZEN10 (44 log CFU/g) and between group C and group ZEN15 (47 log CFU/g) on date D1, and similarly to the caecum, *Staphylococcaceae* were not detected on dates D2 and D3. However, ZEN exerted powerful antimicrobial effects on date D1 (see Figure 5B). The decrease in microbial counts (log CFU/g) indirectly suggests that ZEN does not promote the proliferation of *Staphylococcus* bacteria [47]. It appears that ZEN’s biological effects on digesta were similar to those observed in the caecum in at least two aspects.

Microbiological fermentation of RS [31] leads to the production of SCFAs, cell proliferation, acidification and other beneficial changes [53]. In hypoestrogenic gilts, the bacteriostatic effects of ZEN in distal intestinal segments were manifested on successive analytical dates because ZEN biotransformation occurs in the proximal segments of the gastrointestinal tract [13]. Similarly to the caecum, a closed-loop system was created where ZEN decreased microbial counts, which inhibited SCFA synthesis and, consequently, led to dysbiosis in the descending colon [23,57]. However, the absence of clinical symptoms indicates that eubiosis was not significantly compromised.

The counts of *Clostridium difficile* in the descending colon were also highly similar to those noted in the caecum. Significant differences (*p* ≤ 0.01) were observed only on date D2 between group C and groups ZEN10 and ZEN15 (difference 49.5 and 45.5 log CFU/g, respectively) (see Figure 5C). These findings confirm that the growth of *Clostridium difficile* is stimulated proportionally to the applied ZEN dose. According to recent research [61], this opportunistic strain can lead to intestinal inflammations in piglets and grower-finishing pigs and generate substantial losses in commercial farms [60]. Low doses of ZEN could inhibit *m*RNA expression of both nitric oxide synthases, which decreases nitric oxide levels and suppresses inflammatory processes in the digestive tract, in particular the colon. These processes can contribute to the growth of selected gut bacteria [62]. Therefore, exposure to ZEN stimulates intestinal barrier function, enhances nutrient and protein synthesis, and improves the utilization of energy from substrates that are difficult to degrade (RS), while minimizing the harmful consequences of inflammations and subclinical pathological states [33].

Microbial activity was intensified on date D1 relative to the remaining analytical dates, which could be attributed to the fact that the biological (toxic) effects of ZEN are most pronounced in the first seven days of exposure [13]. Functional variations in the gut microbiome indicate that the intestinal system begins to tolerate ZEN on successive days of exposure [10].

#### 2.3.3. Mycobiome Analysis in Different Intestinal Segments

This is the first study to demonstrate significant differences in yeast and mould counts in the caecum (see Figure 4C,D). Proximal segments of the intestinal tract were also characterised by variations in the counts of mycobiome components, but the noted differences were not statistically significant and were observed only in the experimental groups. In the caecum, *Candida krusei* counts differed significantly on all analytical dates. Yeasts were not detected in group C, which could suggest that ZEN stimulates the growth of yeasts in the digesta. Yeasts were determined on all analytical dates only in group ZEN15 (19.4 to 21.4 log CFU/g) (see Figure 4C). Yeast counts tended to decrease in the remaining experimental groups. In the descending colon, *Candida krusei* counts were very low in group C (see Figure 5D), and this yeast species was not detected on date D3 (average counts were determined at 0.1, 0.2 and 0.0 log CFU/g on successive dates). The abundance of *Candida krusei* was also low in groups ZEN5 and ZEN10. In group ZEN15, *Candida krusei* counts were relatively high on successive dates (12.6, 18.0 and 11.4 log CFU/g, respectively). Significant differences were observed only between dates D2 and D3 in group ZEN15.

Interestingly, only *Fusarium* spp. was detected in the caecum in group C on date D1 (see Figure 4D). Significant differences (*p* ≤ 0.05 and *p* ≤ 0.01) were observed between group C vs. groups ZEN5 and ZEN15 on date D1.

These results imply that both opportunistic mycobiota can cause intestinal mucosa infections, but unlike in other bodily tissues and systems, the difference between fungal overgrowth and fungal infection is difficult to capture [63,64].

#### 2.3.4. Changes in Microbiome and Mycobiome under Exposure to ZEN (Variability of Microbiota)

This subsection analyses quantitative changes in all groups on a given analytical date, in all segments of intestinal tract.

Under exposure to ZEN, significant differences in the counts of five out of the six analysed microbiota were observed on date D1 (see Figure 6). The variations in the abundance of *Escherichia coli* (see Figure 6A) and *Enterococcus faecalis* (see Figure 6B) were similar in all groups during the experiment.

The counts of *Escherichia coli* and *Enterococcus faecalis* (log CFU/g) were low in the first three segments of the intestine. In the caecum and the descending colon, the respective values were two or three times higher, and the differences were statistically significant (*p* ≤ 0.05 and *p* ≤ 0.01). The above findings suggest that ZEN has no effect on *Escherichia coli* or *Enterococcus faecalis*. The dysbiosis index suggests that the biological activity of *Escherichia coli* decreased in the experimental groups relative to group C, but a gradual increase in activity was noted in distal segments of the intestine. According to Youssef and Kamphues [55], *Escherichia coli* counts decrease in response to enhanced fermentation processes and increasing acidification of intestinal digesta. In contrast, the dysbiosis index of *Enterococcus faecalis* was equal to (eubiosis) or higher than 1.0 (dysbiosis) in nearly all intestinal segments (excluding the jejunum) in group ZEN5. The value of the dysbiosis index reached 3.0 in the duodenal cap. These results indicate that ZEN creates a supportive environment for the proliferation of *Enterococcus faecalis* in the first, fourth and fifth segment of the intestine with low pH values [23].

Similar observations were made in metabolomic research [6] which demonstrated that ZEN’s initial stimulatory effect in gilts is neutralised over time by compensatory or adaptive mechanisms, which leads to considerable energy and protein loss in the metabolome. These findings can be used to formulate two hypotheses: (i) feed conversion is more effective, and it contributes to detoxification processes (biotransformation), and (ii) body weight gains increase even under exposure to a MABEL dose. These hypotheses suggest that exposure to low ZEN doses leads to the initiation of compensatory and/or adaptive mechanisms. However, these processes require substantial amounts of energy [55], and they are significantly influenced by the gut microbiome [23].

Similar variations in the counts of coagulase-negative *Staphylococci* were noted in groups C and ZEN5 (see Figure 6C). Significant differences were found between the analysed segments of the duodenum, caecum and the descending colon. In the remaining groups, bacterial counts (log CFU/g) were very low in all intestinal segments, and significant differences were not observed. Interestingly, the values of the dysbiosis index reached 1.0 (eubiosis) in group ZEN5, and were significantly higher than 1.0 in the first three intestinal segments. These findings could suggest that unlike the remaining doses, the MABEL dose stimulates the evaluated bacteria.

On date D1, *Candida krusei* was not detected in any intestinal segments in group C or in the first two segments in group ZEN15. For this reason, the dysbiosis index could not be calculated. On the remaining analytical dates, bacterial counts were higher in the experimental groups by up to 24 log CFU/g, in particular in two distal sampling sites. The above could indicate that ZEN stimulates the proliferation of yeasts in the colon, which may pose health risks to the host. Exposure to ZEN is probably negatively correlated with a diet rich in amino acids, fatty acids and proteins [64]. *Fusarium* spp. counts were also low (see Figure 6E), but in group C, moulds were detected only in the last two intestinal segments. In those segments, the dysbiosis index in group ZEN5 exceeded 3.0, which is indicative of dysbiosis and points to a strong synergistic interaction between ZEN and *Fusarium* spp. in the caecum and a somewhat weaker response in the descending colon.

On date D2, weaker interactions were noted between ZEN and *Escherichia coli* (see Figure 7A) and *Enterococcus faecalis* (see Figure 7B). In the caecum and descending colon, the respective bacterial counts decreased from 60 and 65 log CFU/g to 40 and 45 log CFU/g. Significant differences were also observed between the first three and the last two intestinal segments. The dysbiosis index of *Escherichia coli* in the duodenum increased by 1.0, in particular in groups C/ZEN10 and C/ZEN15. The said increase is indicative of a decrease in pH, which can occur in the initial stages of carbohydrate fermentation [53]. The dysbiosis index of *Enterococcus faecalis* decreased from more than 1.0 on date D1 to below 1.0 on date D2. On date D2, *Clostridium difficile* (see Figure 7C): (i) was not detected in group C; (ii) was detected in the colon at 6 log CFU/g in group ZEN5 and at 42-50 CFU/g in groups ZEN10 and ZEN15. The noted differences were statistically significant, but the dysbiosis index could not be determined in group C. However, it could be hypothesised that LOAEL/NOAEL doses are capable of stimulating the proliferation of *Clostridium difficile* and could exert opposite effects than those suggested in other studies where the examined ZEN doses were several times higher [65]. Our findings indicate that low ZEN doses, in particular NOAEL and smaller doses, exert antibiotic-like effects [29,31]. The above is consistent with the low dose hypothesis [10], whereby high ZEN doses exert toxic effects, whereas low doses can stimulate the development of the macroorganism as well as organisms that do not recognise its presence [66]. *Fusarium* spp. was not detected in group C (see Figure 7D), and the relevant dysbiosis index could not be calculated. In groups ZEN5 and ZEN15, mould counts increased from 0 to 12 log CFU/g (synergistic interaction) with the passage of digesta into more distal intestinal segments. In group ZEN10, *Fusarium* spp. was identified only in the jejunum at 6 log CFU/g.

On date D3, significant functional variations in the microbiome were noted only in *Escherichia coli* (see Figure 8A), *Enterococcus faecalis* (see Figure 8B) and *Clostridium difficile* (see Figure 8C). The microbiome was stabilised relative to dates D1 and D2. *Escherichia coli* counts decreased by approximately 10 log CFU/g on date D2 and were maintained at a similar level on date D3 (see Figure 8A). A different pattern of functional variations was noted in group ZEN5 where microbial counts in the duodenal cap were nearly 40 log CFU/g higher relative to other groups and dates. In the remaining tissues and groups, functional variations were similar to the previous dates, and microbial counts (CFU/g) increased considerably in the caecum and descending colon. Considerable differences were also observed in the values of the dysbiosis index which increased to 2.0 or even 2.5 in group ZEN5 and decreased to 1.0 or less in group ZEN10. Similar results were noted in group ZEN 15, and the highest value of the dysbiosis index (1.5) was noted in the caecum. The functional variations in *Enterococcus faecalis* (see Figure 8B) were nearly identical to those noted on date D2 and similar to those observed in group C. Functional differences were noted between the small intestine and the colon, which can probably be attributed to a local increase in bacterial fermentation [52]. In group ZEN5, the dysbiosis index in the duodenum increased considerably (to 0.65 on date D2 and 2.0 on date D3) relative to group ZEN10 and, partly, in group ZEN15 where similar values (below 1.0) were noted in the colon on date D2. On date D3, the presence of *Clostridium difficile* (see Figure 8C) was observed already in the duodenal cap, whereas on date D2, the caecum was the first intestinal segment colonised by the above microbiota. *Clostridium difficile* was not detected in group C, and its dysbiosis index could not be calculated. These findings suggest that ZEN could promote the development of this bacterial group [31].

#### 2.3.5. Summary

The following observations were made in the intestinal tract of animals exposed to low doses of ZEN in feed: (i)Differences in the dose effect) and variability of microbiota (differences in microbial counts, log CFU/g), mainly in *Escherichia coli* and *Enterococcus faecalis*, were noted between the proximal (microbial counts were lowest in the duodenum) and the distal (microbial counts were highest in the colon) segments of the intestinal tract;(ii)The smallest differences in microbial counts (log CFU/g) were observed in group ZEN5, in particular in proximal intestinal segments (values of the dysbiosis index);(iii)The counts of coagulase-negative *Staphylococci* decreased significantly over time in the evaluated intestinal segments, and these microbiotas were not detected in the colon;(iv)In analyses of the variability of microbiota, *Clostridium difficile* colonies were not identified in group C, but they were detected already in the jejunum and in more distal segments of the intestines in the experimental groups, and microbial counts increased rapidly with an increase in ZEN dose on successive analytical dates (the lowest increase was noted in group ZEN5);(v)ZEN affected the colony counts of intestinal microbiota rather than microbiome diversity, and its effect was greatest in groups ZEN10 and ZEN15. Microbial colony counts were similar in groups ZEN5 and C.

In the analysed mycobiome, ZEN exerted a stimulatory effect on the log values of yeast and mould counts in all intestinal segments, in particular in the colon, and the greatest increase was noted on the first analytical date. Yeast and mould colonies were not detected in group C, excluding on date D1 when they were detected from the jejunum to the descending colon.

The results of this study and our previous findings suggest that the MABEL dose could exert preventive and stimulatory effects on pre-pubertal gilts.

## 3. Materials and Methods

### 3.1. In Vivo Study

#### 3.1.1. General Information

All experimental procedures involving animals were carried out in compliance with Polish regulations setting forth the terms and conditions of animal experimentation (Opinions No. 12/2016 and 45/2016/DLZ of the Local Ethics Committee for Animal Experimentation of 27 April 2016 and 30 November 2016) [5,6].

#### 3.1.2. Experimental Animals and Feed

The in vivo experiment was performed at the Department of Veterinary Prevention and Feed Hygiene of the Faculty of Veterinary Medicine at the University of Warmia and Mazury in Olsztyn on 60 clinically healthy pre-pubertal gilts (grower-finisher crossbred pigs) with initial BW (body weight) of 14.5 ± 2 kg. The animals were housed in pens with free access to water. All groups of gilts received the same feed throughout the experiment. They were randomly assigned to three experimental groups (group ZEN5, group ZEN10 and group ZEN15; *n* = 15) and a control group (group C; *n* = 15) [5,6,67,68]. Group ZEN5 gilts were orally administered ZEN (Sigma-Aldrich Z2125-26MG, Saint Louis, MO, USA) at 5 μg ZEN/kg BW, group ZEN10 pigs—10 μg ZEN/kg BW, and group ZEN15 pigs—15 μg ZEN/kg BW. Analytical samples of ZEN were dissolved in 96 µL of 96% ethanol (SWW 2442-90, Polskie Odczynniki SA, Poland) in weight-appropriate doses. Feed containing different doses of ZEN in an alcohol solution was placed in gel capsules. The capsules were stored at room temperature before administration in order to evaporate the alcohol. In the experimental groups, ZEN was administered daily in gel capsules before morning feeding. The animals were weighed at weekly intervals, and the results were used to adjust individual mycotoxin doses. Feed was the carrier, and group C pigs were administered the same gel capsules, but without mycotoxins [5,6].

The feed administered to all experimental animals was supplied by the same producer. Friable feed was provided ad libitum twice daily, at 8:00 a.m. and 5:00 p.m., throughout the experiment. The composition of the complete diet, as declared by the manufacturer, is presented in Table 1 [5,6].

The proximate chemical composition of diets fed to pigs in groups C, ZEN5, ZEN10, and ZEN15 was determined using the NIRS™ DS2500 F feed analyser (FOSS, Hillerød, Denmark), a monochromator-based NIR reflectance and transflectance analyser with a scanning range of 850–2500 nm [5,6].

#### 3.1.3. Toxicological Analysis

Feed was analysed for the presence of ZEN and DON by high-performance liquid chromatography with UV—vis detection (HPLC-UV). The obtained values did not exceed the limits of quantitation (LOQ) of 2 ng/g for ZEN and 5 ng/g for DON based on the validation of chromatographic methods for the determination of ZEN and DON levels in feed materials and feeds, which was performed at the Department [5,6,69]. This study investigated ZEN and DON which are the most ubiquitous feed contaminants that enter into synergistic interactions [4,18,70,71].

### 3.2. In Vitro Study

#### 3.2.1. Sampling for in Vitro Tests

Five pre-pubertal gilts from every group were euthanized on analytical date 1 (D1—exposure day 7), date 2 (D2—exposure day 21) and date 3 (D3—exposure day 42) by the intravenous administration of pentobarbital sodium (Fatro, Ozzano Emilia BO, Italy) and bleeding. Samples were collected from the gastrointestinal tract of pre-pubertal gilts immediately after cardiac arrest and were prepared for analyses [5,6]. Samples for in vitro analysis were collected from a 10-cm-long intestinal fragment resected from different intestinal segments. Resected fragments were cut from: the duodenum—the first portion (duodenal cup) and the horizontal or third portion; the jejunum, ileum and from the descending colon—from the middle portion. Intestinal segments were tied at both ends before resection to avoid tissue damage. The studied material consisted of sterile digesta samples which were delivered to the microbiological laboratory under refrigerated conditions. A total of 300 samples were collected (20 gilts × 5 intestinal segments × 3 sampling dates).

#### 3.2.2. Microbiological Tests

Microbiological tests were performed at the Microbiological Laboratory of the Non-Public Health Care Centre in Olsztyn, Poland.

Samples of intestinal contents were analysed microbiologically by the culture method according to the relevant ISO standards (PN-EN ISO 18416:2009; PN-EN ISO 21149:2009; PN-EN ISO 22718:2010; PN-EN ISO 22717:2010; PN-EN ISO 21150:2010; PN-EN ISO 16212:2011).

Bacteriological tests. Members of the family *Enterobacteriaceae*, including *Escherichia coli*, *Citrobacter freundii* and *Klebsiella pneumoniae*, were cultured on Biomerieux^®^-France Mac Conkey Agar (MCK) at 37 °C for 24–48 h. *Yersinia enterocolitica* was cultured on Biomerieux^®^-France Yersinia agar (CIN) at 20–25 °C for 3–4 days. *Salmonella* spp. were cultured on Biomerieux^®^-France SS Agar (SS) at 37 °C for 24–48 h. Members of the family *Enterococcaceae*, including faecal strepotococci (Enterococcus), and family *Staphylococcaceae*, including coagulase-negative staphylococci, were cultured on Biomerieux^®^-France Columbia ANC agar + 5% sheep blood (CNA) at 37 °C for 24–48 h. *Clostridium difficile* of the family *Clostridiaceae* was cultured on Biomerieux^®^-France Schaedler agar + 5% sheep blood (SCS) at 37 °C for 48 h.

Fungal tests. *Candida krusei* of the family *Saccharomycetaceae* was cultured on Chrom Agar Candida with a chromogenic mixture, peptone and chloramphenicol (GRASO-Poland) at 37 °C for up to 10 days. *Fusarium* spp. moulds of the family *Nectriaceae* were cultured on Biomerieux^®^-France Dermatophyte agar (DERM) at 20–25 °C for 3–4 weeks.

#### 3.2.3. Microbial Identification

The isolated pure bacterial and fungal cultures were identified in the VITEK 2 system for microbial identification and antibiotic susceptibility testing (Biomerieux^®^, bioMérieux, Craponne, France). Bacteria were additionally identified with the use of standard culture methods based on their morphological features, Gram staining, oxidase production (OXI Gel Polish, Puławy, Poland, Diagnostics Slovakia), coagulase production (Staphytect Plus, Oxoid, Thermo Fisher Microbiology Sale, Hampshire, UK) and the Salmonella Latex agglutination test (Biomex-Kraków, Kraków, Poland). Yeasts were identified based on macroscopic and microscopic characteristics in the VITEK 2 system (Biomerieux^®^-France). Identification tests were performed for the prevalent microflora.

#### 3.2.4. Evaluation of Dysbiosis

The dysbiosis of intestinal microflora was evaluated to determine quantitative differences in bacterial counts between intestinal segments. The dimensionless dysbiosis index was determined based on the ratio of bacterial counts (log CFU/g) in group C and the experimental groups. Values equal to 1.0 were regarded as normal, and values higher or lower than 1.0 were indicative of dysbiosis.

#### 3.2.5. Statistical Analysis

Changes in the log CFU/g values of different bacteria in different sections of the intestine in group C were evaluated under exposure to three doses of ZEN. Data were obtained on three analytical dates, and they were processed separately for each date. The log CFU/g values of each type of bacteria were divided into groups based on two factors: (a) ZEN dose, and (b) the analysed intestinal segment (variability of microbiota). Two-way ANOVA could not be performed because bacteria were not detected in all groups or log CFU/g values were identical (zero variance). Therefore, the following analyses were carried out: (i) differences in log CFU/g values in the same intestinal segment in the control group were determined under exposure to three doses of ZEN, and (ii) differences in log CFU/g values in different intestinal segments in the control group were determined under exposure to the same dose of ZEN. In both scenarios, the observed differences between groups (1—ZEN dose/2—section of the intestine) were processed by one-way ANOVA. Differences between pairs of means were determined with Tukey’s multiple comparison test. If no bacterial colonies were observed or if all log CFU values were identical in any of the compared groups, one-way ANOVA was performed for the remaining groups, and group means were compared against zero or against the value of an excluded group with the use of Student’s *t*-test. Data were processed in Statistica v. 13 (TIBCO Software Inc., 2017, Warsaw, Poland).

## Figures and Tables

**Figure 1 toxins-11-00296-f001:**
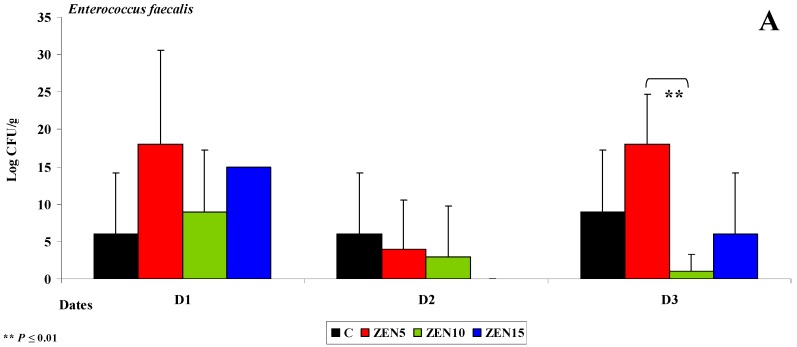
The dose effect of ZEN on functional diversity in the microbiome in the duodenal cap: (**A**,**B**) arithmetic means ($\bar{x}$) and standard deviation (SD) in five samples collected on each analytical date (D1, D2 and D3) in the evaluated groups (C, ZEN5, ZEN10 and ZEN15); Statistically significant differences: * at *p* ≤ 0.05 and ** *p* ≤ 0.01.

**Figure 2 toxins-11-00296-f002:**
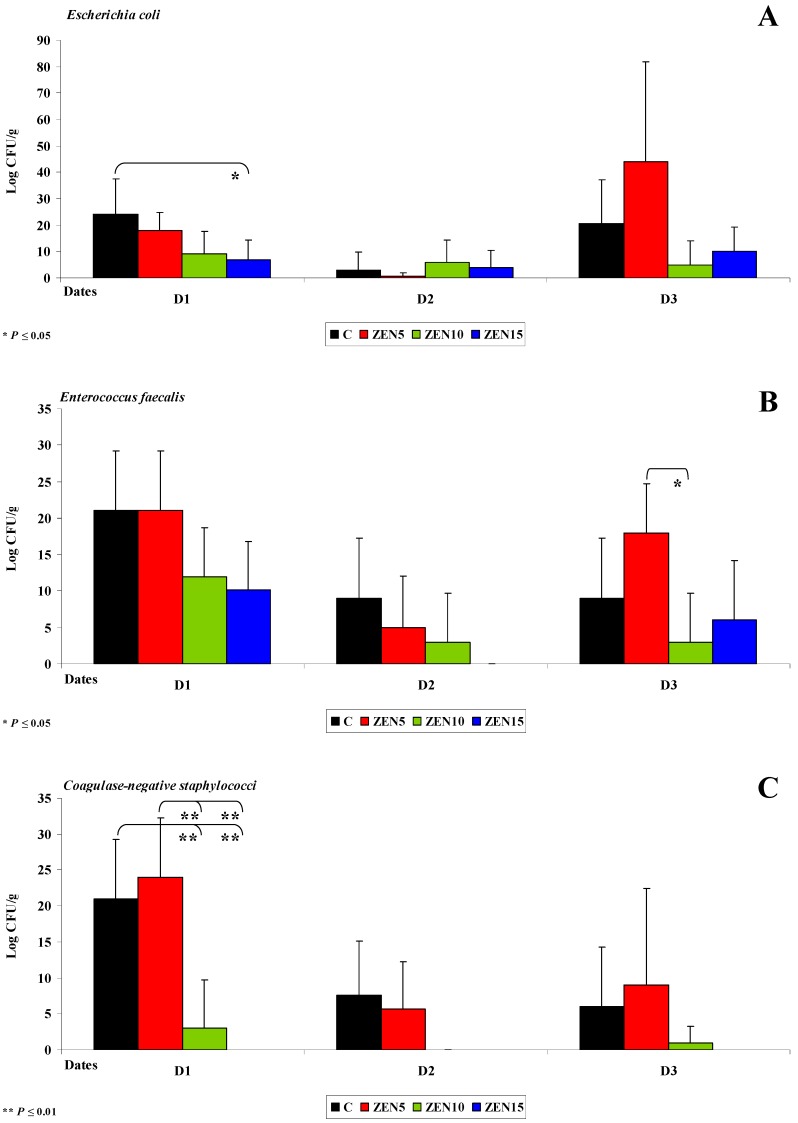
The dose effect of ZEN on functional diversity in the microbiome in the third part of duodenum: (**A**–**C**) Arithmetic means ($\bar{x}$) and standard deviation (SD) in five samples collected on each analytical date (D1, D2 and D3) in the evaluated groups (C, ZEN5, ZEN10 and ZEN15). Statistically significant differences: * at *p* ≤ 0.05 and ** *p* ≤ 0.01.

**Figure 3 toxins-11-00296-f003:**
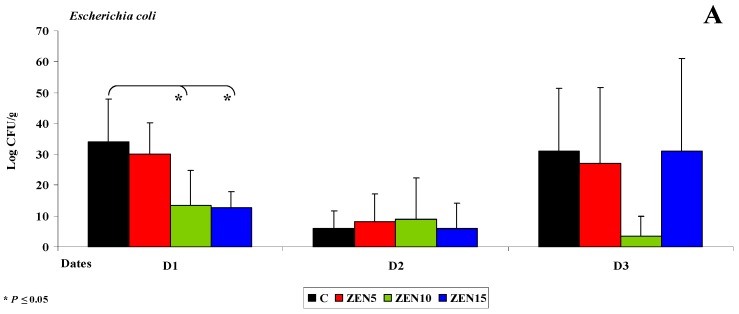
The dose effect of ZEN on functional diversity in the jejunal microbiome: (**A**–**D**) arithmetic means ($\bar{x}$) and standard deviation (SD) in five samples collected on each analytical date (D1, D2 and D3) in the evaluated groups (C, ZEN5, ZEN10 and ZEN15). Statistically significant differences: * at *p* ≤ 0.05 and ** *p* ≤ 0.01.

**Figure 4 toxins-11-00296-f004:**
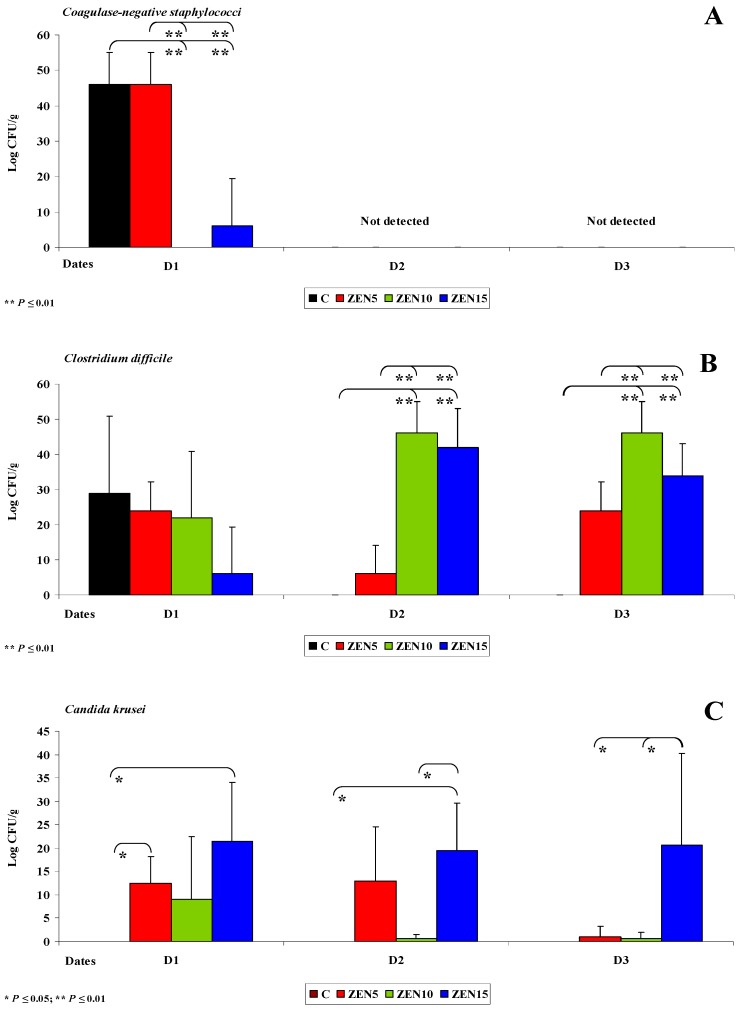
The dose effect of ZEN on functional diversity in the caecal microbiome: (**A**–**D**) arithmetic means ($\bar{x}$) and standard deviation (SD) in five samples collected on each analytical date (D1, D2 and D3) in the evaluated groups (C, ZEN5, ZEN10 and ZEN15). Statistically significant differences: * at *p* ≤ 0.05 and ** *p* ≤ 0.01.

**Figure 5 toxins-11-00296-f005:**
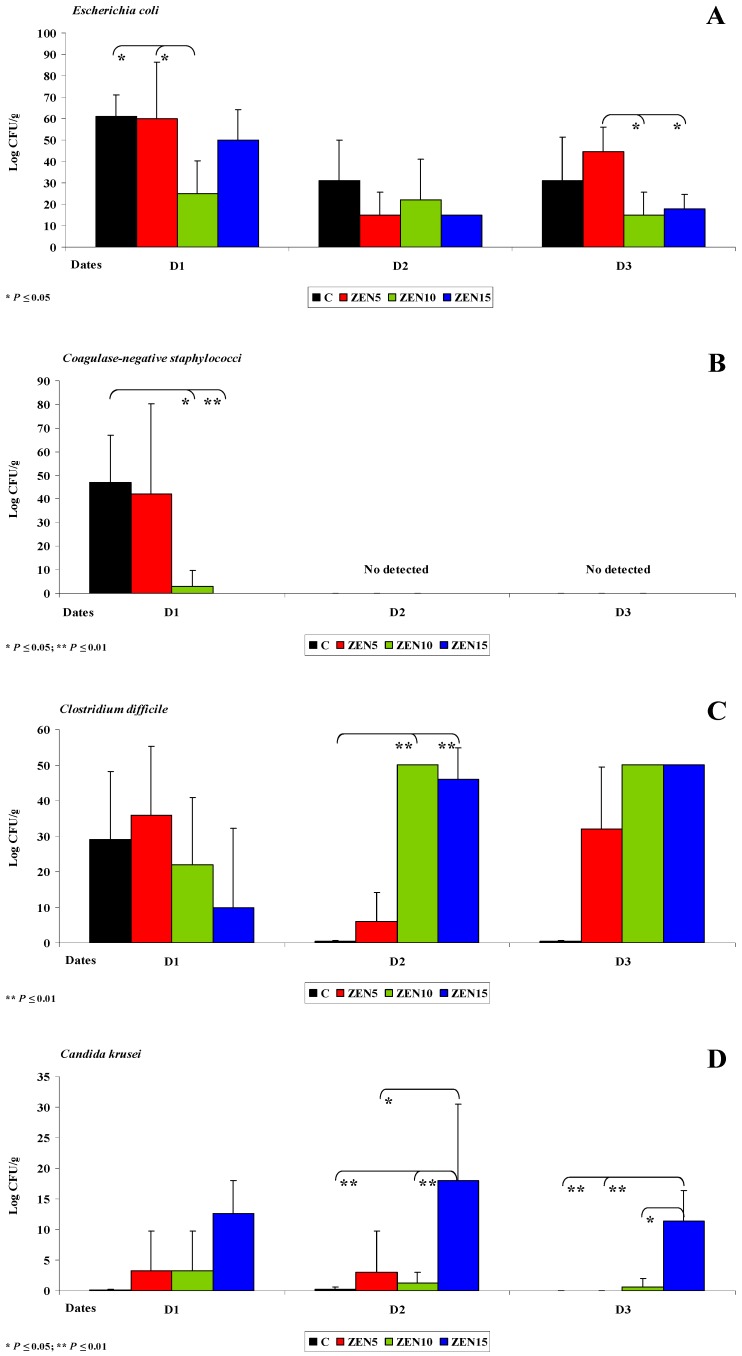
The dose effect of ZEN on functional diversity in the descending colon microbiome: (**A**–**D**) arithmetic means ($\bar{x}$) and standard deviation (SD) in five samples collected on each analytical date (D1, D2 and D3) in the evaluated groups (C, ZEN5, ZEN10 and ZEN15). Statistically significant differences: * at *p* ≤ 0.05 and ** *p* ≤ 0.01.

**Figure 6 toxins-11-00296-f006:**
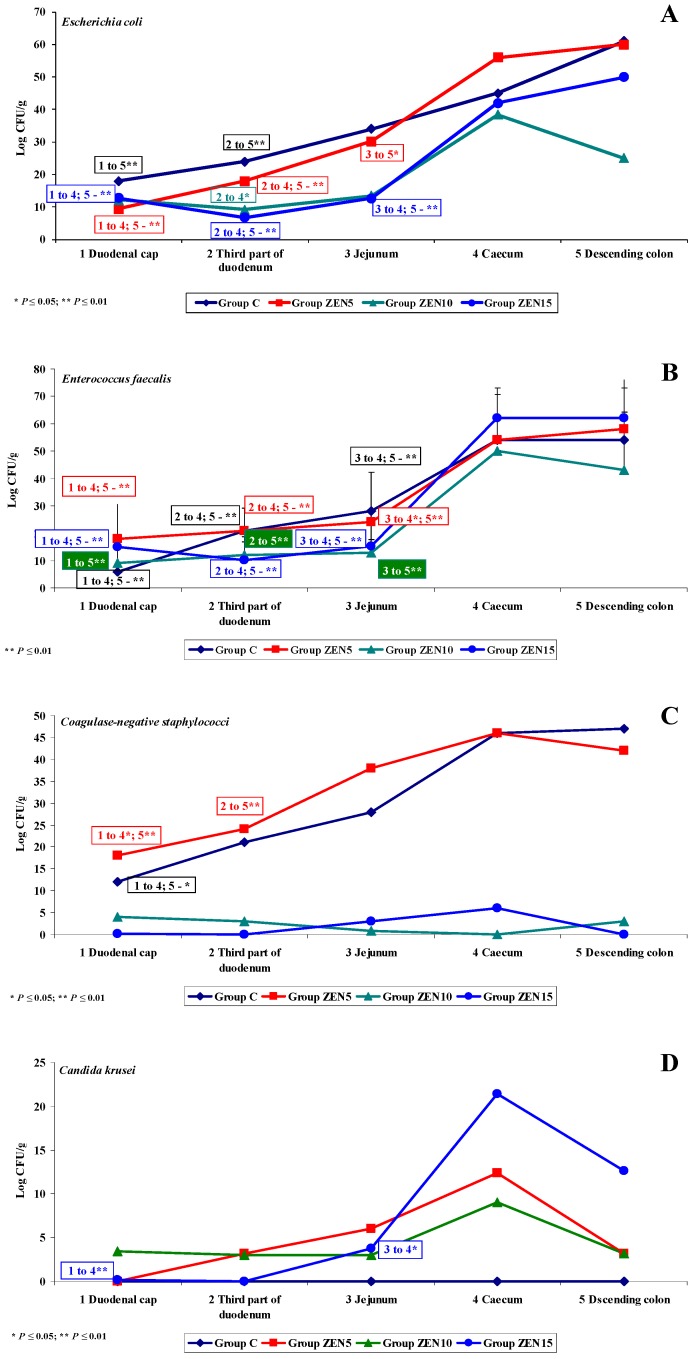
Variability of microbiota. Variations in the counts of selected microbiota and mycobiota under exposure to ZEN on the first analytical date (D1): (**A**–**E**) arithmetic means ($\bar{x}$) in five samples of selected bacterial strains (Groups C, ZEN5, ZEN10 and ZEN15). Statistically significant differences: * at *p* ≤ 0.05 and ** *p* ≤ 0.01.

**Figure 7 toxins-11-00296-f007:**
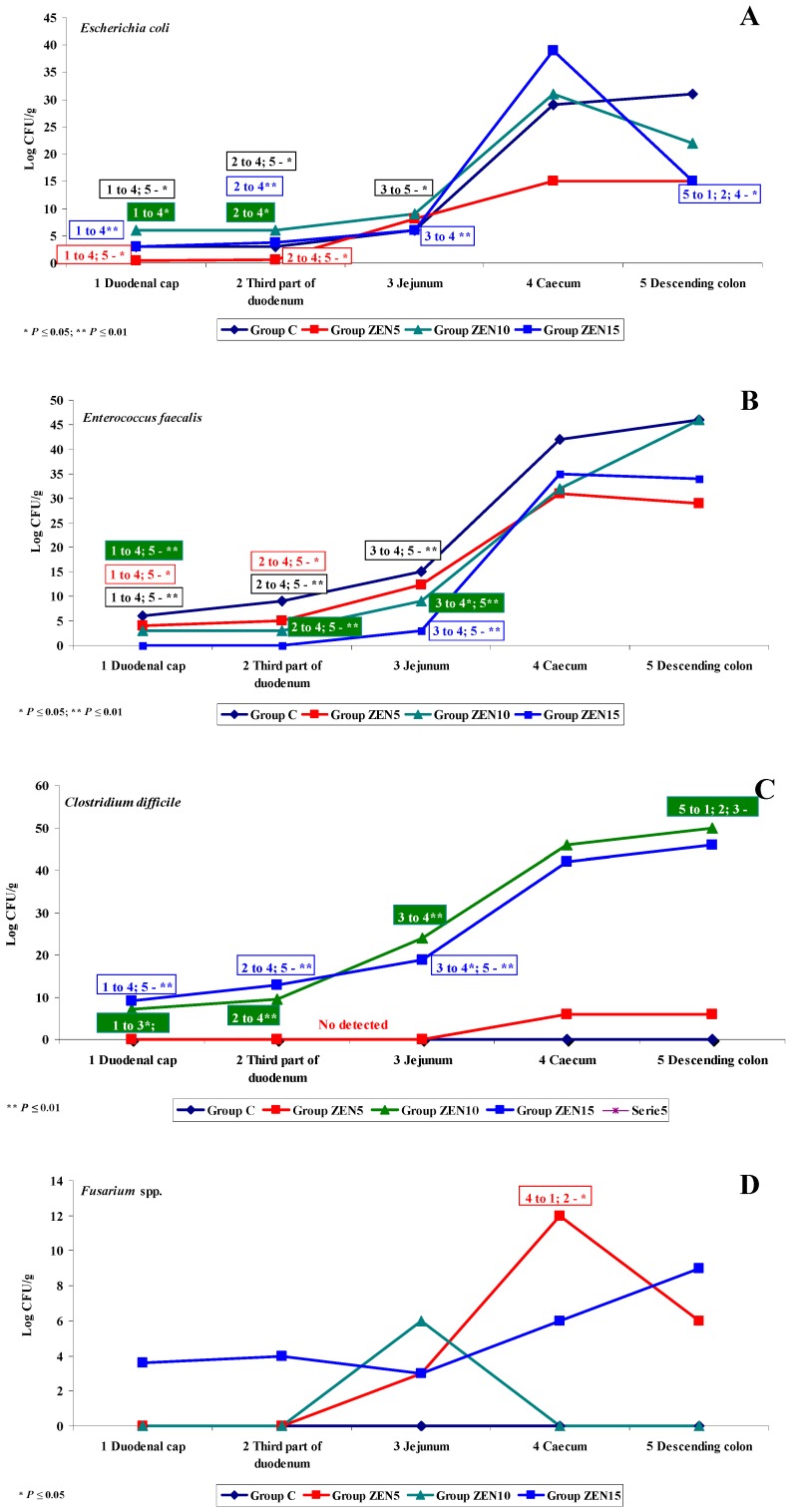
Variability of microbiota. The effect of ZEN on functional diversity in selected microbiota and mycobiota on analytical date D2: (**A**–**D**) arithmetic means ($\bar{x}$) in five samples of selected bacterial strains on (Groups C, ZEN5, ZEN10 and ZEN15). Statistically significant differences: * at *p* ≤ 0.05 and ** *p* ≤ 0.01.

**Figure 8 toxins-11-00296-f008:**
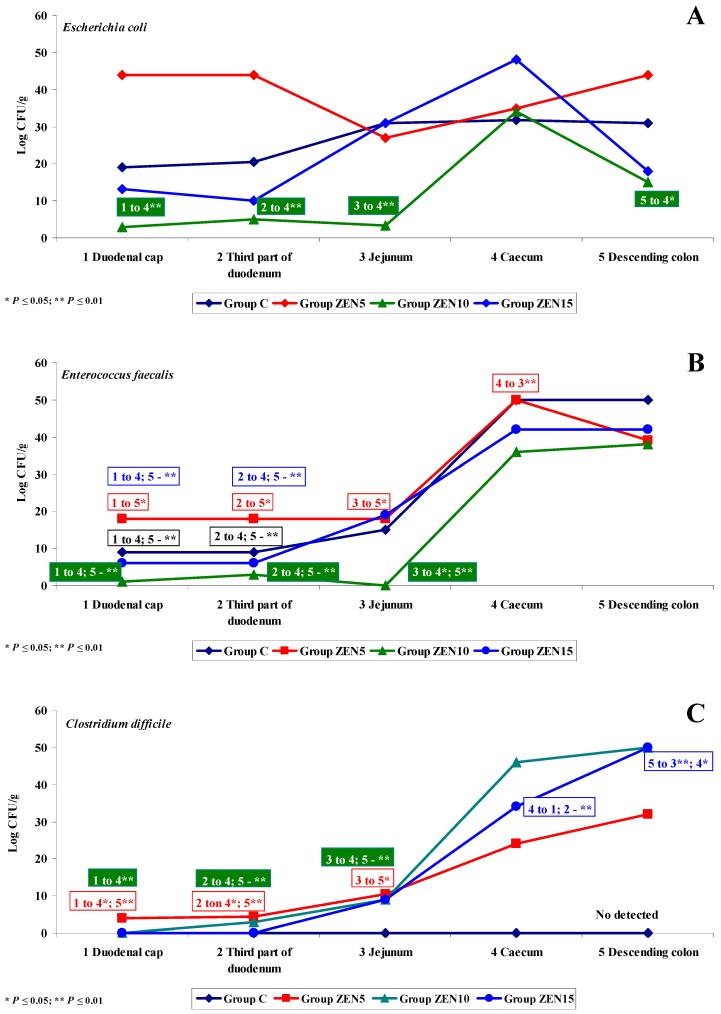
Variability of microbiota. The effect of ZEN on functional diversity in the microbiome on analytical date D3: (**A**–**C**) arithmetic means ($\bar{x}$) in five samples of selected bacterial strains (Groups C, ZEN5, ZEN10 and ZEN15). Statistically significant differences: * at *p* ≤ 0.05 and ** *p* ≤ 0.01.

**Table 1 toxins-11-00296-t001:** Declared composition of the complete diet.

Parameters	Composition Declared by the Manufacturer (%)
Soybean meal	16
Wheat	55
Barley	22
Wheat bran	4.0
Chalk	0.3
Zitrosan	0.2
Vitamin-mineral premix ^1^	2.5

^1^ Composition of the vitamin-mineral premix per kg: vitamin A—500.000 IU; iron—5000 mg; vitamin D3—100.000 IU; zinc—5000 mg; vitamin E (alpha-tocopherol)—2000 mg; manganese—3000 mg; vitamin K—150 mg; copper (CuSO_4_·5H_2_O)—500 mg; vitamin B_1_—100 mg; cobalt—20 mg; vitamin B_2_—300 mg; iodine—40 mg; vitamin B_6_—150 mg; selenium—15 mg; vitamin B_12_—1500 μg; L-lysine—9.4 g; niacin—1200 mg; DL-methionine+cystine—3.7 g; pantothenic acid—600 mg; L-threonine—2.3 g; folic acid—50 mg; tryptophan—1.1 g; biotin—7500 μg; phytase+choline—10 g; ToyoCerin probiotic+calcium—250 g; antioxidant+mineral phosphorus and released phosphorus—60 g; magnesium—5 g; sodium and calcium—51 g.

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
