# Peer review of "Time-Dependent Changes in the Intestinal Microbiome of Gilts Exposed to Low Zearalenone Doses"

_toxins, 2019, doi:10.3390/toxins11050296_

Round 1
Reviewer 1 Report
Abstract
Line 12, replace the word “mould” with “Fusarium species fungi”
Introduction
-Replace the reference number 1 with another appropriate one.
-Line 38, what do you mean by “pure” … toxic effect of the mycotoxins is not “recent research”, this is established from many decades. Please rewrite the whole sentence.
-line 64, replace the word “produce” with “exert”
-line 65, replace the word surprising with another appropriate word.
-line 76, 77, 78… add reference(s)
Results and Discussion
-line 90, there is a debate for the word masked and modified mycotoxins. Some researchers used interchangeably and other differentiate between them . in your case I would use the word modified and omit the second one.
-line 97, these are the results ? I see only some literature review , please clarify the importance on this part in section results and discussion.
Author Response
Abstract
Line 12, replace the word “mould” with “Fusarium species fungi”
The suggested correction has been made.
Introduction
-Replace the reference number 1 with another appropriate one.
The indicated reference has been replaced with:
Montanha, F.Z.; Anater, A.; Burchard, J.F.; Luciano, F.B.; Meca, G.; Manyes, L.; Pimpão, C.T. Mycotoxins in dry-cured meats: A review. Food Chem. Toxicol. 2018, 111, 494-502. https://doi.org/10.1016/j.fct.2017.12.008
-Line 38, what do you mean by “pure” … toxic effect of the mycotoxins is not “recent research”, this is established from many decades. Please rewrite the whole sentence.
The sentence has been rewritten as follows: “However, extensive research conducted in the last decade indicates that health problems resulting from exposure to small doses of the parental compound [4-6] without modified mycotoxins [7] can be equally important”.
-line 64, replace the word “produce” with “exert”
The suggested correction has been made.
-line 65, replace the word surprising with another appropriate word.
The word “surprising” has been replaced with “different”
-line 76, 77, 78… add reference(s)
The following reference has been added:
Sun, X.; Jia, Z. Microbiome modulates intestinal homeostasis against inflammatory diseases. Vet. Immunol. Immunop. 2018, 205, 97-105. https://doi.org/10.1016/j.vetimm.2018.10.014
Results and Discussion
-line 90, there is a debate for the word masked and modified mycotoxins. Some researchers used interchangeably and other differentiate between them. In your case I would use the word modified and omit the second one.
The word “masked” has been deleted.
-line 97, these are the results ? I see only some literature review, please clarify the importance on this part in section results and discussion.
The relevant fragment has been moved to the Introduction section.
Reviewer 2 Report
General comment
This manuscript has interest but it is very difficult to read and follow.
Authors should make a statistical analysis of the results that agrees with the experimental design. It seems that 3 factors are studied: dose of ZEN, time of analysis, and portion of the digestive tract on which the bacterial counts are measured. Only a statistic analysis of the dose was done. In my opinion that is not correct. (see also detailed comments L365-473)
Authors used 4 different kind of Figures to present the same data, which is not acceptable. In my opinion the best way to present the data is the one used in Fig10. It is together the most demonstrative and the most easy to read. I suggest to keep only this kind of presentation. Text should be completely rewritten in consequence. (see also detailed comments L166-193)
Authors should discuss the method used to measure effects of ZEN (ZON preferred?) on the microbiote regarding other methods available (16S mRNA analysis).
Also, a brief chapter in the introduction on the toxicokinetic of ZEN seems necessary as it could explain why different effect could be observed in the different portion of the gut (concentration of ZEN in the lumen varies because of its absorption).
Specific comments
L18-20 : clarify how ZEN was obtained and how it was administered. If necessary, what was given to the control group?
L21-28: it is difficult to understand the results of this study by reading this. Is the effect statistically significant? Is the authors observed something like a “dose response” or a “time effect” ? The protocol described L13-20 suggests that the purpose of the study was to investigate a dose-effect (3 doses) and a time-effect (3 times), results should be written in agreement.
L34-51: out of subject here, should be removed
L52-70: too wide, focus on ZEN, its known effects in pig, and maximum tolerated level in feed.
L71-83: too wide, should focus on 1- what are the methods to analyze the gut microbiote (gold standards), 2-what is known about the effect of mycotoxins at low doses on the microbiote. Detailed information are expected in this introduction to explain your protocol.
L88-90: A discussion of the mycotoxin analyzed is necessary. Why only DON and ZON?
L97-108: not necessary
L110: A discussion of the method used to investigate the microbiote is necessary, specially because of known variability in CFU.
L128-153: I do not understand what the message is? How is it possible to say “Enterococcus faecalis were fairly stable … throughout the experiment” and “considerable differences were observed between groups”. Purpose of the study seems to analyze a dose effect and a time effect, Fig1A and 1B seem done to reveal that, so what is the conclusion?
Fig 1C and 1D: why these figure? What groups were used, what means “selected groups”, you pooled data in all the groups, in controls, in ZEN ?
L154-158: not necessary
L166-193: same comment, I do not understand the analysis of data made. What means “A significant decrease in the counts (log CFU/g) of … was noted in groups ZEN10 and ZEN15 on all analytical dates relative to group C”? Is the difference observed on each day of measure or from control on D1? How statistical analysis is presented in the Fig cannot help to understand: it is more common to have asterisk on the different group rather than a text box.
About statistical analysis by itself, Fig2B, E faecalis not detectable in ZEN15 on D2 but not statistically different from C? Also, Staph were not detectable on D2 in ZEN10 and ZEN15 but there was no statistical difference between groups? Same comment on D3 for ZEN15.
All one way ANOVA is not appropriated to analyze results, at least 2 factor ANOVA should be done: one factor is the dose, another factor is the time.
Fig 3: As it is not clear what the used data to make Fig3 are, I do not understand this Figure.
L193: it is very uncommon to read “ZEN has bacteriostatic or even bactericidal effects”, these words should not be used to effects related to the microbiote.
L194-364: same comments as L166-193
L365-473: These figures show the same data, you have to make a choice in the way you present your results. This choice should be dictated by the main scientific question you are asking and it should impose the statistical analysis you are conducting. What your main objective is:
- analyzing a dose effect?
- analyzing an effect of time?
- analyzing an effect of the digestive tract segment?
Once this choice has been made, you decide on a statistical analysis and a single method of presenting the results.
Fig10A: Probably the best presentation of results (add error bars). I suggest to keep this one and remove all the others.
Fig11: why this figure? It is confusing and difficult to read.
L474-495: need rewriting to highlight the most important effects of the 3 factors studied: dose, time, portion of the gut.
L535-540: in my opinion this analysis of feed is not enough to guarantee the absence of other mycotoxins.
L554-581: introduction or discussion necessary on this methods regarding 16S mRNA analysis
L588-589: your protocol reveals you have 3 doses at 3 times of observation for 5 portions of the gut. In my opinion one way ANOVA is not appropriated to this analysis!
Author Response
General comment
This manuscript has interest but it is very difficult to read and follow.
Authors should make a statistical analysis of the results that agrees with the experimental design. It seems that 3 factors are studied: dose of ZEN, time of analysis, and portion of the digestive tract on which the bacterial counts are measured. Only a statistic analysis of the dose was done. In my opinion that is not correct. (see also detailed comments L365-473)
Unfortunately, we cannot fully agree with the Reviewer because each of the studied factors was subjected to a separate statistical analysis (L-597), and the results are discussed at the end of the manuscript (L596-602).
The experiment analysed the same animals and the same samples that were collected post mortem. The samples were subjected to qualitative and quantitative tests, and the results were presented graphically from different points of view. The first group of charts presents the dose effect on different analytical dates in a given intestinal segment (bar charts). The second group of charts presents the variability in microbiota and/or mycobiota on different analytical dates in all evaluated intestinal segments. Therefore, the obtained data were subjected to separate statistical analyses (different methodological approaches) with the use of the same method. The results would be completely illegible if they were presented in a single chart.
Authors used 4 different kind of Figures to present the same data, which is not acceptable. In my opinion the best way to present the data is the one used in Fig10. It is together the most demonstrative and the most easy to read. I suggest to keep only this kind of presentation. Text should be completely rewritten in consequence. (see also detailed comments L166-193)
The second and fourth type of figures (i.e. Figures 3, 5, 7, 9 and 11) have been deleted.
Authors should discuss the method used to measure effects of ZEN (ZON preferred?) on the microbiote regarding other methods available (16S mRNA analysis).
The last paragraph of the Introduction section clearly states that the effect of low doses of ZEN on microbial counts was determined with the use of conventional analytical methods which are most frequently deployed in microbiological studies. A new paragraph describing other microbiological methods has been included in the Introduction in the revised manuscript.
Also, a brief chapter in the introduction on the toxicokinetic of ZEN seems necessary as it could explain why different effect could be observed in the different portion of the gut (concentration of ZEN in the lumen varies because of its absorption).
To our knowledge, the toxicokinetics of ZEN at such low doses has never been studied in relation to the microbiome, even in phenotypic analyses. The referenced studies [4, 5, 6, 10, 13, 18] discuss factors that could indirectly influence the microbiome (L52-83). The results of our study analysing the toxicokinetics of ZEN are still being processed, and they will probably be published in 2020.
Specific comments
L18-20 : clarify how ZEN was obtained and how it was administered. If necessary, what was given to the control group?
The method of obtaining, administering and dosing ZEN was presented in subsection 3.1.2 “Experimental animals and feed”. The administration procedure in the control (reference) group was described in lines 525-526.
L21-28: it is difficult to understand the results of this study by reading this. Is the effect statistically significant? Is the authors observed something like a “dose response” or a “time effect” ? The protocol described L13-20 suggests that the purpose of the study was to investigate a dose-effect (3 doses) and a time-effect (3 times), results should be written in agreement.
The relevant fragment of the Abstract has been rewritten as follows: “Static (determined on a given day of the experiment) and dynamic (determined throughout the experiment) differences in microbial counts (log CFU/g), in particular Escherichia coli and Enterococcus faecalis, were observed between the proximal and the distal duodenum”.
L34-51: out of subject here, should be removed
If possible, we would prefer to keep this fragment in the text because it justifies the purpose of the experiment. This pioneering study investigates the effect of very low doses of ZEN which are often ingested by animals and which exert unknown effects bordering on the physiology and pathology of microorganisms and macroorganisms. Mammals have learned to tolerate low doses of mycotoxins which, in some situations, are essential for animal (macroorganism) health.
L52-70: too wide, focus on ZEN, its known effects in pig, and maximum tolerated level in feed.
The entire paragraph focuses on ZEN, and it cites the most recent review articles and research studies concerning the gastrointestinal tract. It discusses different types of low doses (LOAEL, NOAEL and MABEL) which are in no way related to the maximum tolerated level of ZEN in feed.
L71-83: too wide, should focus on 1- what are the methods to analyze the gut microbiote (gold standards), 2-what is known about the effect of mycotoxins at low doses on the microbiote. Detailed information are expected in this introduction to explain your protocol.
The effect of very low doses of ZEN on the porcine gut microbiota has never been studied, and this paper presents the results of a pioneering experiment. The interactions between the gut microbiome and much higher doses of ZEN have been investigated previously in a referenced study [4]. In the revised manuscript, the methods applied to analyse the gut microbiome are described in a separate paragraph at the end of the Introduction section.
L88-90: A discussion of the mycotoxin analyzed is necessary. Why only DON and ZON?
According to the results of studies monitoring mycotoxin levels around the world, animal feeds are contaminated predominantly with doxynivalenol and zearalenone. The relevant information has been added to subsection 3.1.3: “This study investigated ZEN and DON which are the most ubiquitous feed contaminants that enter into synergistic interactions”.
L97-108: not necessary
The relevant fragment has been moved to the Introduction section.
L110: A discussion of the method used to investigate the microbiote is necessary, specially because of known variability in CFU.
The following text has been added to subsection 2.3.1:
I – The overgrowth of the small intestinal microbiome, including changes in microbial counts and/or microbial types, was not observed. In the proximal segment of the small intestine, the counts of non-pathogenic bacterial strains exceeded 105 log CFU due to colonisation by bacterial strains that are ubiquitous in the colon.
II – Only the results where significant differences were noted are presented in the figures.
L128-153:
I do not understand what the message is? How is it possible to say
“Enterococcus faecalis were fairly stable … throughout the experiment”
and “considerable differences were observed between groups”. Purpose of
the study seems to analyze a dose effect and a time effect, Fig1A and 1B
seem done to reveal that, so what is the conclusion?
Fig 1C and 1D:
why these figure? What groups were used, what means “selected groups”,
you pooled data in all the groups, in controls, in ZEN ?
The relevant fragment has been modified to highlight the dose effect: “The above is reflected in the fact that the counts of Enterococcus faecalis did not exceed 18 log CFU/g throughout the experiment (see Figure 1A) and were relatively low in comparison with other intestinal segments. However, considerable differences were observed between groups (5 to 18 log CFU/g) on selected analytical dates”.
Figures 1C and 1D have been deleted.
L154-158: not necessary
The relevant fragment has been deleted.
L166-193:
same comment, I do not understand the analysis of data made. What means
“A significant decrease in the counts (log CFU/g) of … was noted in
groups ZEN10 and ZEN15 on all analytical dates relative to group C”? Is
the difference observed on each day of measure or from control on D1?
How statistical analysis is presented in the Fig cannot help to
understand: it is more common to have asterisk on the different group
rather than a text box.
About statistical analysis by itself, Fig2B,
E faecalis not detectable in ZEN15 on D2 but not statistically
different from C? Also, Staph were not detectable on D2 in ZEN10 and
ZEN15 but there was no statistical difference between groups? Same
comment on D3 for ZEN15.
All one way ANOVA is not appropriated to
analyze results, at least 2 factor ANOVA should be done: one factor is
the dose, another factor is the time.
The control (reference) group is described in L517-518. The number of pigs was identical in the control group and in each experimental group, throughout the experiment. Control group animals were administered the same feed in gel capsules, but without the addition of ZEN. Group C is marked in black colour in every figure drawing.
One-way ANOVA was performed (2x), depending on the results of the analyses investigating the dose effect and changes in microbiota.
Fig 3: As it is not clear what the used data to make Fig3 are, I do not understand this Figure.
Figure 3 has been deleted.
L193: it is very uncommon to read “ZEN has bacteriostatic or even bactericidal effects”, these words should not be used to effects related to the microbiote.
This phrase does not relate to the microbiota, but it proposes the assumption that ZEN could have bacteriostatic or bactericidal effects. The above had been discussed in the referenced study [10] as well as by other authors, including Zheng et al. [48] in Toxins. Fusarium fungi are the main producers of ZEN, and similarly to other moulds, they secrete antibiotics and/or mycotoxins with comparable antibacterial effects. Therefore, we believe that the questioned statement is justified.
L194-364: same comments as L166-193
Please refer to the previous explanation.
L365-473:
These figures show the same data, you have to make a choice in the way
you present your results. This choice should be dictated by the main
scientific question you are asking and it should impose the statistical
analysis you are conducting. What your main objective is:
- analyzing a dose effect?
- analyzing an effect of time?
- analyzing an effect of the digestive tract segment?
Once this choice has been made, you decide on a statistical analysis and a single method of presenting the results.
The same data (log CFU/g) were processed twice by ANOVA, but every statistical analysis was carried out to elicit different information, i.e. the dose effect and variations in gut microbiota. Therefore, ANOVA was performed twice to analyse the same data from different points of view, and the results were presented graphically.
Fig10A: Probably the best presentation of results (add error bars). I suggest to keep this one and remove all the others.
In the revised manuscript, the results are presented in two types of charts. The method of presenting statistically significant differences has been changed in the first type of charts.
Fig11: why this figure? It is confusing and difficult to read.
Figure 11 has been deleted.
L474-495: need rewriting to highlight the most important effects of the 3 factors studied: dose, time, portion of the gut.
Every analysed effect in different intestinal segments is briefly described/summarised at the end of every sub-section. The five most characteristic changes induced by ZEN were discussed in the final conclusions. The last sentence sums up our overall findings.
L535-540: in my opinion this analysis of feed is not enough to guarantee the absence of other mycotoxins.
The results presented in the table indicate that feed did not contain factors promoting the growth of fungi or enhancing the activity of the studied mycotoxin. Feed rations supported mycotoxin biotransformation or detoxification in the macroorganism, as previously discussed by Rykaczewska et al. 2018 [6].
L554-581: introduction or discussion necessary on this methods regarding 16S mRNA analysis
The relevant information has been moved to the Introduction section for better readability.
L588-589: your protocol reveals you have 3 doses at 3 times of observation for 5 portions of the gut. In my opinion one way ANOVA is not appropriated to this analysis!
The same data (log CFU/g) were processed twice by ANOVA, but every statistical analysis was carried out to elicit different information, i.e. the dose effect and variations in gut microbiota. Therefore, ANOVA was performed twice to analyse the same data from different points of view. Our expectations were met because each ANOVA produced clear and univocal results.
Reviewer 3 Report
The paper entitled “Time-dependent changes in the intestinal microbiome of gilts exposed to low zearalenone doses” is interesting since it deals with a very important topic: the effect of mycotoxins (Zearalenone, in the present case) intake on the modulation of microorganism population colonizing the intestine of animals (pigs, in the present case). The output of this kind of experimentation are relevant from economic and sanitary point of view. They also provide a report of the state of the art on this topic. In this sense, my only comment is that, as honestly reported by the authors of the paper, many of the data provided seem to be a confirmation of (reinforcing) what they have already reported. However new data on low doses effects on micro- and myco-biome are added and discussed accordingly. The experimental designs are well conducted and most of the data are convincing.
Author Response
The paper entitled “Time-dependent changes in the intestinal microbiome of gilts exposed to low zearalenone doses” is interesting since it deals with a very important topic: the effect of mycotoxins (Zearalenone, in the present case) intake on the modulation of microorganism population colonizing the intestine of animals (pigs, in the present case). The output of this kind of experimentation are relevant from economic and sanitary point of view. They also provide a report of the state of the art on this topic. In this sense, my only comment is that, as honestly reported by the authors of the paper, many of the data provided seem to be a confirmation of (reinforcing) what they have already reported. However new data on low doses effects on micro- and myco-biome are added and discussed accordingly. The experimental designs are well conducted and most of the data are convincing.
Reviewer III gave a very good review - without comments.Round 2
Reviewer 2 Report
General comment
1- Authors have improved their manuscript but this study is still difficult to read and understand because the same results are presented 2 fold and the statistical analysis are unclear or unappropriated:
- Fig 1 to 5 show the effect of dose (what authors called “static”) and effect of time (what authors called “dynamic”), but the stats are not clear. Where are the results of the analysis of the effect of time on these Figures? I cannot find it in the legend nor in the Figure.
- Statistic analysis on Fig 6 to 10 is easier to follow, but what is the rationale of this analysis? Statistical analysis should reveal whether the microbiote profile, which can vary in controls with the part of the intestine tract analyzed, is affected by ZEN exposure or not. A linear model should be used to reveal that information. A comparison of the bacterial CFU in each portion of the digestive tract done separately in each treatment group has very weak interest.
2- Explanation provided by authors about statistical analysis is not clear:
L21, authors said « Static … dynamic (determined throughout the experiment) differences in microbial counts (log CFU/g), in particular Escherichia coli and Enterococcus faecalis, were observed”, but what does that means? Did two way ANOVA was used? Where are the detailed explanations of the stats in M&M section? Where are the results of “dynamic” analysis in text and Figures?
Specific comments
L146-150: I do not understand what authors wanted to say in these lines
Fig 1B and all Fig 1 to 5: sometimes when a bacterial species was not detected there is a box to say that this group is different from other one, sometimes no, why?
Fig 2A: “controls” are in dark blue, while in black in all other Figures
Fig 4A: y axis, “No presence…”?
Fig 6 the text is in green in a white box, while in Fig 7 the text is in white in a green box, why?L462-466: this is not a conclusion
L531-532: really? so it will be easy to add several bibliographic references to strengthen this affirmation of “synergic interactions”
L574-578: Taking into this informations, what is the rationale of a statistic analysis based on numeric values?
Author Response
General comment
1- Authors have improved their manuscript but this study is still difficult to read and understand because the same results are presented 2 fold and the statistical analysis are unclear or unappropriated:
The same results are presented twice, but from two different points of view. The first presentation concerns differences observed between analytical dates (D1, D2 and D3) (quantitative changes, the analysed microbiota, differences between groups, the analysed intestinal segments), i.e. the changes induced by the analysed doses of ZEN.
The second presentation concerns the variability of (quantitative changes in) microbiota in the gastrointestinal system on different analytical dates (dynamic changes that take place in the entire digestive system during the experiment), which supported a statistical comparison of microbial parameters (numerical values) in the evaluated segments of the digestive tract on different analytical dates.
The relevant corrections have been made in the manuscript:
L 153: This subsection analyses the effects of the applied ZEN doses on quantitative changes in microbiota in different groups on different analytical dates, in the same segment of the gastrointestinal tract.
L 357: This subsection analyses quantitative changes in microbiota in all groups on a given analytical date, in all segments of the gastrointestinal tract.
We hope that the introduced changes will improve the manuscript’s readability.
- Fig 1 to 5 show the effect of dose (what authors called “static”) and effect of time (what authors called “dynamic”), but the stats are not clear. Where are the results of the analysis of the effect of time on these Figures? I cannot find it in the legend nor in the Figure.
Static effects refer to different segments of the intestinal tract, whereas dynamic effects denote differences in microbial counts (log CFU) in the entire intestinal tract.
To avoid confusion, the term “static” has been replaced with “dose effect”, and the term “dynamic” – with “variability of microbiota”. Figure and table legends have been modified accordingly.
- Statistic analysis on Fig 6 to 10 is easier to follow, but what is the rationale of this analysis? Statistical analysis should reveal whether the microbiote profile, which can vary in controls with the part of the intestine tract analyzed, is affected by ZEN exposure or not. A linear model should be used to reveal that information. A comparison of the bacterial CFU in each portion of the digestive tract done separately in each treatment group has very weak interest.
The aim of the experiment was to evaluate the effect of various ZEN doses on the microbial profile in different segments of the intestinal tract. The results of all statistical analyses cannot be presented comprehensively in graphic form. They have been presented in two stages: (i) in Figures 1-5 which illustrate the effect of various ZEN doses on different microbiota colonizing selected segments of the intestinal tract in groups on all analytical dates; and (ii) in Figures 6-8 which compare the counts (log CFU) of selected microbiota in groups on different analytical dates.
The results generated by both approaches were summarized in subsection 2.3.5. which has been renamed from “Conclusions” to “Summary” in the revised manuscript.
2- Explanation provided by authors about statistical analysis is not clear:
L21, authors said « Static … dynamic (determined throughout the experiment) differences in microbial counts (log CFU/g), in particular Escherichia coli and Enterococcus faecalis, were observed”, but what does that means? Did two way ANOVA was used? Where are the detailed explanations of the stats in M&M section? Where are the results of “dynamic” analysis in text and Figures?
L21-23 – The following explanations have been provided in the revised manuscript:
Differences in the log values of microbial counts, mainly Escherichia coli and Enterococcus faecalis, were observed between the proximal and distal segments of the intestinal tract on different analytical dates as well as in the entire intestinal tract.
In the revised manuscript, the presentation of statistical results in subsection 3.2.5 of the M&M section has been modified as follows:
Changes in the log CFU/g values of different bacteria in different sections of the intestine in the control group were evaluated under exposure to three doses of ZEN. Data were obtained on three analytical dates, and they were processed separately for each date. The log CFU/g values of each type of bacteria were divided into groups based on two factors: (a) ZEN dose, and (b) the analysed intestinal segment. Two-way ANOVA could not be performed because bacteria were not detected in all groups or log CFU/g values were identical (zero variance). Therefore, the following analyses were carried out: (i) differences in log CFU/g values in the same intestinal segment in the control group were determined under exposure to three doses of ZEN, and (ii) differences in log CFU/g values in different intestinal segments in the control group were determined under exposure to the same dose of ZEN. In both scenarios, the observed differences between groups (1 - ZEN dose/ 2 - section of the intestine) were processed by one-way ANOVA. Differences between pairs of means were determined with Tukey’s multiple comparison test. If no bacterial colonies were observed or if all log CFU values were identical in any of the compared groups, one-way ANOVA was performed for the remaining groups, and group means were compared against zero or against the value of an excluded group with the use of Student’s t-test. Data were processed in Statistica v. 13 (TIBCO Software Inc., 2017).
In response to the Reviewer’s first question, we have proposed to replace the term “dynamic” with “variability of microbiota”.
Specific comments
L146-150: I do not understand what authors wanted to say in these lines
These lines provide additional evidence that the obtained results were not indicative of pathological changes. This fragment was incorrectly positioned in the text, and it has been relocated in the revised manuscript.
Fig 1B and all Fig 1 to 5: sometimes when a bacterial species was not detected there is a box to say that this group is different from other one, sometimes no, why?
Bacterial species that were not detected and not presented in Figures 1 to 5 are described in detail in subsection 2.3.1. Some cases were not described due to the absence of statistical differences or because a given microbiome was not identified. In other cases, log CFU/g values were very low (not visible in figure drawings, such as Figure 2C).
Fig 2A: “controls” are in dark blue, while in black in all other Figures
The relevant correction has been made.
Fig 4A: y axis, “No presence…”?
The relevant correction has been made.
Fig
6 the text is in green in a white box, while in Fig 7 the text is in
white in a green box, why?L462-466: this is not a conclusion
The relevant correction has been made.
L531-532:
really? so it will be easy to add several bibliographic references to
strengthen this affirmation of “synergic interactions”
The appropriate references have been added.
L574-578: Taking into this informations, what is the rationale of a statistic analysis based on numeric values?
Please note that statistical results are used mainly in scientific research, whereas the dysbiosis index is relevant mostly for clinicians.
Dysbiosis is a general concept that includes pathological changes induced by non-physiological composition of intestinal microflora. Dysbiosis is closely correlated with immune, digestive and metabolic functions. Intestinal inflammations such as ulcerative colitis are regarded as manifestations of dysbiosis, where pathologically changed intestinal microflora leads to unnecessary stimulation of the intestinal immune system. Dysbiosis requires dietary modifications and, if diagnostically confirmed, pharmacological treatment to regulate metabolic and immune processes.
Healthy intestinal function requires a certain number of “good” gut bacteria. From the medical point of view, the counts of potentially pathogenic bacteria and fungi have to be limited. Ideally, the intestinal system should be in a state of a complete microbial balance. Obviously, not all gut bacteria can be identified and the appropriate norms cannot be set for all microorganisms. However, based on the existing knowledge, we can analyse selected microbial genera/species with confirmed beneficial (protective and immunostimulating microflora that nourishes the intestinal epithelium) as well as pathogenic or potentially pathogenic effects. Most importantly, a healthy microbial balance in the intestine cannot be achieved simply by eliminating pathogenic bacteria (although this is necessary in some cases, such as in infections caused by Clostridium difficile), but by enhancing the growth of beneficial bacteria that inhibit the proliferation of pathogenic bacteria and fungi.